# Distinct patterns of within-host virus populations between two subgroups of human respiratory syncytial virus

Gu-Lung Lin [1,2✉], Simon B. Drysdale [1,2,27], Matthew D. Snape[1,2], Daniel O'Connor [1,2], Anthony Brown [3], George MacIntyre-Cockett[4], Esther Mellado-Gomez[4], Mariateresa de Cesare [4], David Bonsall[4,5], M. Azim Ansari[4], Deniz Öner[6], Jeroen Aerssens[6], Christopher Butler[7], Louis Bont[8,9], Peter Openshaw [10], Federico Martinón-Torres [11,12], Harish Nair [13], Rory Bowden [4,28], RESCEU Investigators*, Tanya Golubchik [5,29] & Andrew J. Pollard [1,2,29]

Human respiratory syncytial virus (RSV) is a major cause of lower respiratory tract infection in young children globally, but little is known about within-host RSV diversity. Here, we characterised within-host RSV populations using deep-sequencing data from 319 nasopharyngeal swabs collected during 2017–2020. RSV-B had lower consensus diversity than RSV-A at the population level, while exhibiting greater within-host diversity. Two RSV-B consensus sequences had an amino acid alteration (K68N) in the fusion (F) protein, which has been associated with reduced susceptibility to nirsevimab (MEDI8897), a novel RSV monoclonal antibody under development. In addition, several minor variants were identified in the antigenic sites of the F protein, one of which may confer resistance to palivizumab, the only licensed RSV monoclonal antibody. The differences in within-host virus populations emphasise the importance of monitoring for vaccine efficacy and may help to explain the different prevalences of monoclonal antibody-escape mutants between the two subgroups.

[1] Oxford Vaccine Group, Department of Paediatrics, University of Oxford, Oxford, UK. [2] NIHR Oxford Biomedical Research Centre, Oxford, UK. [3] Peter Medawar Building for Pathogen Research, University of Oxford, Oxford, UK. [4] Wellcome Centre for Human Genetics, University of Oxford, Oxford, UK. [5] Big Data Institute, Nuffield Department of Medicine, University of Oxford, Oxford, UK. [6] Translational Biomarkers, Infectious Diseases Therapeutic Area, Janssen Pharmaceutica NV, Beerse, Belgium. [7] Nuffield Department of Primary Care Health Sciences, University of Oxford, Oxford, UK. [8] Department of Pediatrics, Wilhelmina Children's Hospital, University Medical Center Utrecht, Utrecht, Netherlands. [9] ReSViNET Foundation, Zeist, Netherlands. [10] National Heart and Lung Institute, Imperial College London, London, UK. [11] Translational Pediatrics and Infectious Diseases, Hospital Clínico Universitario de Santiago de Compostela, Santiago de Compostela, Spain. [12] Genetics, Vaccines, Infectious Diseases, and Pediatrics Research Group (GENVIP), Instituto de Investigación Sanitaria de Santiago de Compostela, Santiago de Compostela, Spain. [13] Centre for Global Health, Usher Institute, Edinburgh Medical School, University of Edinburgh, Edinburgh, UK. [27] Present address: Paediatric Infectious Diseases Research Group, Institute for Infection and Immunity, St George's, University of London, London, UK. [28] Present address: Division of Advanced Technology and Biology, Walter and Eliza Hall Institute of Medical Research, Melbourne, VIC, Australia. [29] These authors jointly supervised this work: Tanya Golubchik, Andrew J Pollard. *A list of authors and their affiliations appears at the end of the paper. ✉email: gu-lung.lin@paediatrics.ox.ac.uk

Human respiratory syncytial virus (RSV) is the leading cause of lower respiratory tract infection (LRTI) in young children, globally responsible for around 33 million episodes of LRTI in children under 5 years of age annually with a disproportionately high burden in infants younger than 1 year of age[1]. Repeated infection is common throughout life[2], usually resulting in mild symptoms, but it can also cause serious disease in older (age ≥65 years) or immunocompromised adults and people with chronic cardiopulmonary disease[3]. Despite decades of effort, there is no efficacious antiviral for treatment or licensed vaccine to prevent RSV infection, and thus the standard of care is supportive management only. Palivizumab, an RSV-specific humanised monoclonal antibody, is the only available immunoprophylactic agent. It requires multiple administrations over the RSV season and is very expensive, so its use is limited to the highest-risk populations, namely infants born preterm and those with congenital heart disease, chronic pulmonary disorders or severe combined immunodeficiency[4].

RSV is a negative-sense single-stranded RNA virus with a genome containing ten genes. The F gene encodes the fusion (F) glycoprotein, which mediates the fusion of host cell and viral membranes. The F protein is the main target for antibody-mediated neutralisation, and has been the focus of the development of vaccines and monoclonal antibodies[5]. Through the fusion process, the F protein changes from the prefusion to postfusion conformation. Several antigenic sites (neutralising epitopes in particular) have been located on the surface of the F protein. Antibodies exclusively targeting prefusion-specific antigenic sites (e.g. sites ∅ and V) are more potent than those targeting sites that can be found in both conformations (e.g. sites I, II, IV)[6]. Nirsevimab (MEDI8897), a recombinant human monoclonal antibody currently in phase 3 clinical trials, exclusively targets antigenic site ∅[7], and suptavumab (REGN2222), another prefusion-specific monoclonal antibody, binds antigenic site V[8]. Palivizumab and its affinity-enhanced variant, motavizumab[9], target antigenic site II, and antibody 101F binds antigenic site IV[10]. Mutations in the antigenic sites that confer resistance to monoclonal antibodies have been identified. For example, mutants with N262S/Y, N268I, K272E/N/M/T/Q or S275F/L in the F protein are less susceptible to palivizumab[11–13], and nirsevimab has reduced neutralising activity against mutants with N67I/N208Y, N208S/D, K68N/N201S or K68N/N208S in the F protein[7].

The G gene encodes the attachment (G) glycoprotein, a transmembrane protein responsible for viral attachment. The extracellular portion (ectodomain) of the G protein consists of two hypervariable mucin-like regions flanking a conserved central domain (CCD)[14]. The CCD, containing antigenic sites γ1 and γ2, has been shown to be a target for neutralising antibodies[15] and is another focus of vaccine development[16,17]. Outside the CCD, the mucin-like regions also have multiple antigenic sites though less well-defined[18]. The mucin-like region II (second hypervariable region) has been shown to have hypermutation at the population level and has thus been used widely in phylogenetic analyses[19].

The two subgroups of RSV (A and B) co-circulate in epidemics, and both exhibit rapid evolutionary dynamics[20]. Molecular epidemiology and evolutionary dynamics of RSV have been extensively studied at the consensus level; however, little is known about virus populations in each infected individual (i.e. within-host or intrahost virus diversity). Using high-throughput whole-genome sequencing, it is now possible to sequence viruses in sufficient depth to obtain a complete picture of within-host populations. A previous study showed that within-host RSV diversity increased in an immunocompromised infant with persistent RSV infection following a haematopoietic stem cell transplant, and palivizumab escape mutants emerged after multiple administrations of this drug[21]. Another study demonstrated that RSV-A exhibited greater within-host virus diversity in experimentally infected adults than in naturally infected infants[22]. However, these results were limited to RSV-A infection and did not look at natural infections in adult populations. Analysing within-host virus genetic diversity in infections that represent general seasonal epidemics can aid understanding of the patterns of virus evolution and its driving forces, informing the development of preventative and treatment measures.

In this study, we seek to characterise within-host RSV populations for the two subgroups, RSV-A and RSV-B, using deep sequencing of samples collected from participants in three prospective clinical studies. We find that RSV-B exhibits greater within-host diversity than RSV-A, with two RSV-B consensus strains and one RSV-B minor variant likely conferring resistance to nirsevimab or palivizumab. We also show that temporal changes of intrahost viral populations follow stochastic patterns. Our work highlights the importance of continued genetic surveillance of RSV to ensure the effectiveness of future RSV vaccines and therapeutics.

## Results

**Sample population**. We sequenced RSV from 858 nasopharyngeal swabs collected from 459 RSV-infected patients in the United Kingdom, Spain and the Netherlands during 2017–2020. Of these, 327 samples had sufficient viral load to generate more than 10,000 unique (deduplicated) RSV reads. After removing five samples containing both RSV-A and RSV-B, 322 samples were included in the within-host virus diversity analysis. Sequencing was carried out in four batches, with 11, 113, 41 and 157 of the included samples from each batch respectively (Supplementary Table 1). The 322 samples were collected from 267 different participants, among which 34 participants had multiple samples (mean 2.6, range 2–5) collected on different days (ranging from 1 to 8 days apart).

**Cumulative minor allele frequencies and minor variants**. Genomic positions with a read depth of less than 200 were excluded from the analysis. Nearly 90% of the samples had ≥80% of the genome passing this threshold. Three samples had a significantly high mean cumulative minor allele frequency (MAF) per sample: 0.52% (from an RSV-A-infected infant; batch 4), 0.19% (from an RSV-B-infected adult; batch 2) and 0.17% (from an RSV-B-infected infant; batch 4). These samples presumably represented a real or artefactual mixture of genetically distinct strains of the same RSV subgroup and were thus excluded from the following analysis. The sources and sequencing yields of the remaining 319 samples (collected from 264 participants) are shown in Table 1.

The median of the mean cumulative MAF per sample was 0.039% (range 0.025–0.068%) for the 319 samples. The distributions of the mean cumulative MAF per sample were significantly different between samples from different sequencing batches (Supplementary Fig. 1a), likely due to the differences in the ratio of duplicate read counts to total RSV read counts (percent duplication rate) between batches (Supplementary Table 1). After adjusting for the observed batch effects (e.g. Supplementary Fig. 1b), RSV-B samples had a higher mean cumulative MAF per sample than RSV-A samples (median of the original data: 0.042% vs. 0.037%; multiple linear regression with batch and the number of unique RSV reads as covariates, $P = 0.016$; Mann–Whitney $U$-test on standardised data, $P = 0.016$).

On average, each sample had 3.7 minor variants (range 0–30; defined as variants with a frequency of ≥3%). Of the samples, 18.8% (60/319) did not have any minor variants. An inverse

**Table 1 Characteristics of RSV samples by subgroup.**

|  | RSV-A ($N = 175$) | RSV-B ($N = 144$) | P value[a] |
|---|---|---|---|
| Host number |  |  | 0.12[b] |
| Infants | 141 | 115 |  |
| Older adults | 1 | 7 |  |
| Host age, median (range) |  |  |  |
| Infants (month)[c] | 4.5 (0.5-11.6) | 4.3 (0.2-11.7) | 0.72 |
| Older adults (year) | 69 | 75 (72-78) | 0.19 |
| Sample source |  |  | 0.45 |
| United Kingdom | 74 | 64 |  |
| Netherlands | 58 | 53 |  |
| Spain | 43 | 27 |  |
| Sampling season |  |  | $2.9 \times 10^{-5}$ |
| 2017–18 | 14 | 33 |  |
| 2018–19 | 65 | 63 |  |
| 2019–20 | 96 | 48 |  |
| Days between symptom onset and sample collection, median (range)[d] | 4 (1-11) | 4 (1-9) | 0.11 |
| Number of unique RSV read pairs ($\log_{10}$), median (range) | 4.6 (4.0-5.8) | 4.7 (4.0-5.9) | 0.22 |
| Batch 1 | 4.9 (4.1-5.5) | 5.3 (4.4-5.6) | 0.50 |
| Batch 2 | 4.6 (4.0-5.6) | 4.6 (4.0-5.9) | 0.98 |
| Batch 3 | 4.4 (4.0-4.8) | 4.5 (4.0-5.5) | 0.18 |
| Batch 4 | 4.7 (4.0-5.8) | 4.9 (4.0-5.6) | 0.12 |
| Minimum genome coverage (%) | 99.9 | 100 | 0.37 |
| Average depth of coverage, median (range) | 3372 (696-7897) | 3650 (525-7930) | 0.41 |
| Batch 1 | 2940 (696-6823) | 4975 (1295-7601) | 0.63 |
| Batch 2 | 3561 (1092-7452) | 3469 (1091-7930) | 0.68 |
| Batch 3 | 2045 (803-3224) | 2258 (525-7157) | 0.37 |
| Batch 4 | 3736 (847-7897) | 4505 (719-7798) | 0.23 |

[a]Unless otherwise specified, chi-square tests with Yates' continuity correction or Fisher's exact tests were used for contingency analysis, and two-tailed Mann–Whitney U-tests were used to compare numeric variables between subgroups.
[b]Logistic regression was used to adjust for sampling season. Samples were collected from older adults only in 2017–18 and 2018–19 RSV seasons, when RSV-B was the predominant circulating subgroup.
[c]One infant with RSV-B infection had missing information on age.
[d]Six infants with RSV-A infection and five infants with RSV-B infection had missing information on the date of symptom onset.

correlation was noted between the number of unique RSV reads and the number of minor variants ($r = -0.41$, $P = 4.2 \times 10^{-14}$; Supplementary Fig. 2), consistent with a greater variance of MAF when the sampling fraction was small (i.e. few unique reads were sequenced)[23]. Variation rarely occurred at the same genomic position in different samples. Among all minor variants found in this study, only 5.9% (57/972) were shared by multiple samples (excluding 17 minor variants only shared by sequential samples from the same participants), usually no more than five samples. However, there was one minor variant shared by 59% (85/144) of the RSV-B samples, with a frequency between 3 and 11%. This minor variant had a G to A substitution at position 3403 of the L gene, causing an amino acid alteration from glutamic acid to lysine at position 1135 (E1135K) of the RNA-dependent RNA polymerase.

**Potential antigenic variants**. The sequences encoding the antigenic sites of the F protein were highly conserved at the consensus level in this study. However, two RSV-B isolates from two infant participants, both of whom had only one sample collected, had an A to T substitution at nucleotide position 204 of the F gene. This substitution results in an amino acid alteration from lysine to asparagine (K68N), which in a previous study was associated with a fourfold reduction in susceptibility to nirsevimab neutralisation in vitro[7]. No minor variant was found at this position in these two samples.

The frequencies and distribution of all minor variants across the coding sequence of the F gene are shown in Fig. 1a. There were one, eight, two and three minor variants identified in the antigenic sites $\varnothing$, II, IV and V of the F protein, respectively (Table 2). 0, 6.0% (6/100) and 1.6% (2/124) of the participants had potential antigenic variants (i.e. minor variants encoding a

nonsynonymous substitution in the antigenic sites) in the 2017–18, 2018–19 and 2019–20 RSV seasons, respectively. One of these minor variants had two nucleotide substitutions with a frequency of ≥3% in a single codon, encoding an amino acid substitution from isoleucine to threonine at position 261 (I261T). Other minor variants identified in the antigenic sites were from different samples. To date, none of these variants have been reported to confer resistance to monoclonal antibodies.

We also looked at the frequencies and distribution of minor variants in the coding region of the G gene (Fig. 1b). The median frequency of minor variants was significantly higher in the G gene than in the F gene, either at potential antigenic sites (median: 9.3% vs. 4.6%; Mann–Whitney U-test, $P = 0.022$) or across the whole coding sequences (median: 8.3% vs. 4.4%; Mann–Whitney U-test, $P = 0.004$), consistent with previous studies identifying the G gene as the most variable gene in the virus genome[14]. The median minor variant frequency in the mucin-like region II of the G gene (13.7%) was greater than that in the mucin-like region I (9.2%), which was greater than that in the CCD (4.0%). However, these differences were not statistically significant (Kruskal–Wallis test, $P = 0.20$).

**Pairwise nucleotide diversity**. Within-host virus genetic diversity was estimated as pairwise nucleotide diversity (see Methods). Pairwise nucleotide diversity did not correlate with the number of unique RSV reads after adjusting for the batch effects (Supplementary Table 2 and Supplementary Fig. 3a), but was highly consistent with the mean cumulative MAF per sample ($r = 0.997$, $P < 2.2 \times 10^{-16}$; Supplementary Fig. 3b). The median pairwise nucleotide diversity of the whole dataset was 0.0007 (range 0.0005–0.0014). Gene-wise comparisons showed that the L gene had significantly higher pairwise nucleotide diversity than the

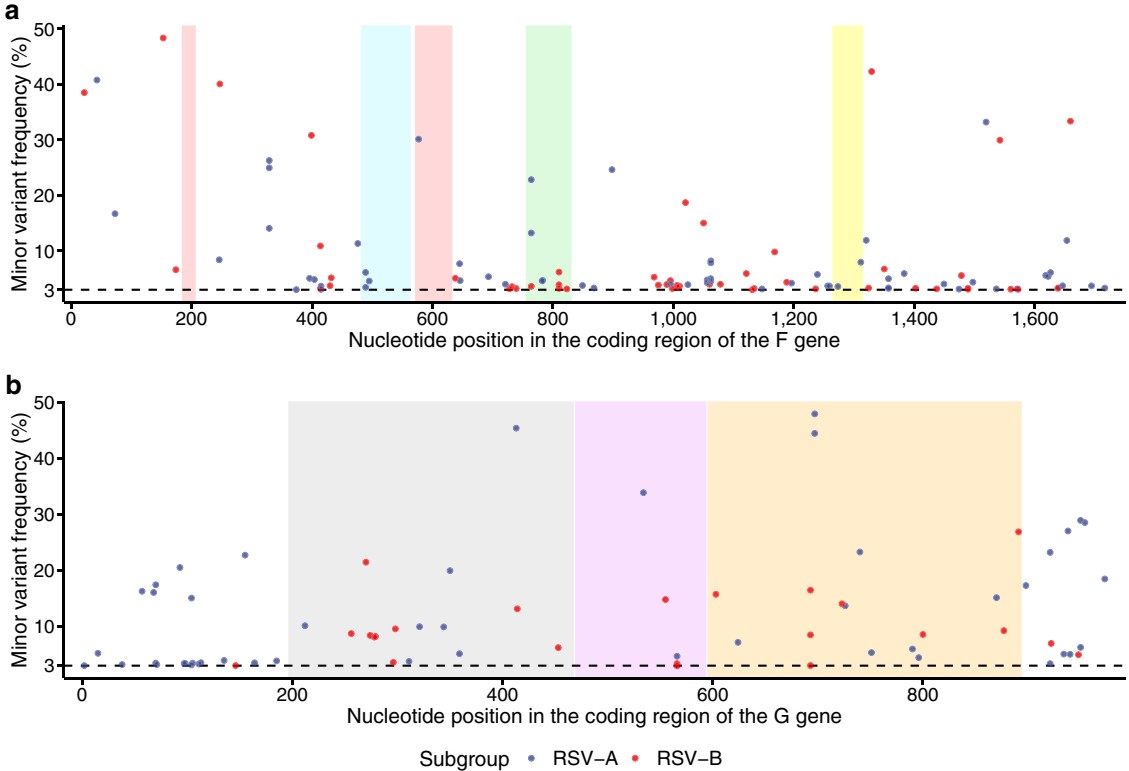

Subgroup   • RSV–A   • RSV–B

**Fig. 1 Minor variants in the coding region of the F and G genes among 175 RSV-A and 144 RSV-B samples. a** F gene. Shaded regions represent known antigenic sites (neutralising epitopes in particular): red, prefusion-specific antigenic site $\varnothing$ (target for nirsevimab); green, site II (target for palivizumab and motavizumab); yellow, site IV (target for 101F) and blue, prefusion-specific site V (target for suptavumab). **b** G gene. The purple region represents the conserved central domain (target for 3D3 and 2D10), flanked by highly variable mucin-like regions I (grey) and II (orange). Nirsevimab, palivizumab, motavizumab, 101F, suptavumab, 3D3 and 2D10 are RSV-specific monoclonal antibodies. Each dot denotes a minor variant, coloured by subgroup. Black dashed line represents minor allele frequency of 3%, used to define a minor variant. Positions are numbered from the first base of the coding sequence of each gene according to the NCBI reference sequence (accession number NC_038235).

| Table 2 Characteristics of minor variants within the antigenic sites of the fusion protein. | | | | |
|---|---|---|---|---|
| **Nucleotide position[a]** | **Codon change** | **Amino acid change[b]** | **Antigenic site** | **Subgroup/country/season/ minor allele frequency (%)[c]** |
| 489 | GAA:GAt | E163D | V | A/GB/2018–19/3.4 |
| | | | | A/GB/2018–19/6.1 |
| 495 | AAC:AAt | N165 | V | A/GB/2018–19/4.6 |
| 577 | CCA:tCA[d] | P193S | $\varnothing$ | A/GB/2018–19/30.1 |
| 764 | AGT:AaT | S255N | II | A/ES/2019–20/13.2[e] |
| | | | | A/ES/2019–20/22.8[e] |
| | | | | B/GB/2018–19/3.6 |
| 782 | ATC:Act | I261T | II | A/ES/2018–19/4.6[f] |
| 783 | ATC:Act | I261T | II | A/ES/2018–19/4.7[f] |
| 810 | CAG:CAa | Q270 | II | B/NL/2017–18/3.2 |
| | | | | B/NL/2017–18/6.2[g] |
| | | | | B/NL/2018–19/3.9 |
| 823 | TCA:cCA | S275P | II | B/GB/2018–19/3.1 |
| 1273 | TCA:cCA | S425P | IV | A/GB/2019–20/3.6 |
| 1311 | AAC:AAt | N437 | IV | A/NL/2019–20/8.0 |

[a]Positions are numbered from the first base of the coding sequence of the F gene according to the NCBI reference sequence (accession number NC_038235).
[b]Positions are numbered from the first methionine of the fusion protein according to the NCBI reference sequence (accession number NC_038235).
[c]GB denotes the United Kingdom; ES, Spain and NL, the Netherlands.
[d]55.7% (98/176) of the RSV-A samples had a consensus base of T, and all RSV-B samples had a consensus base of T at this position.
[e]These two variants were found in samples collected from the same participant on day 2 (13.2%) and day 5 (22.8%) of hospitalisation, respectively. Samples collected from this participant on other days (days 1, 3 and 4) did not have variants with a frequency of ≥3% at this position.
[f]These two were co-occurring mutations, identified in the same minor variant.
[g]Except for this variant, which was in a sample from an adult participant, other minor variants were identified in infant samples.

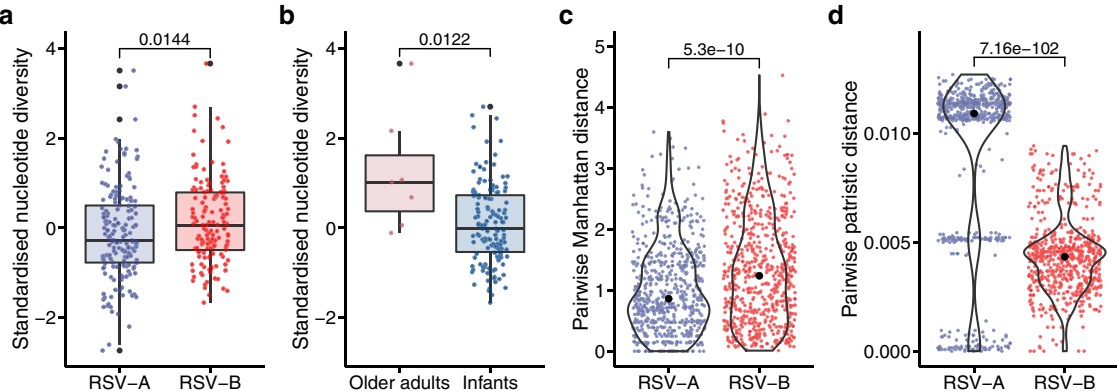

**Fig. 2 Z-score standardised pairwise nucleotide diversity and pairwise genetic distances. a** Comparison of standardised pairwise nucleotide diversity between 175 RSV-A and 144 RSV-B samples. **b** Comparison of standardised pairwise nucleotide diversity of RSV-B between seven adult samples and 137 infant samples. RSV-A isolates were excluded from this comparison because only one adult had RSV-A infection. **c** Comparison of pairwise Manhattan distances. **d** Comparison of pairwise patristic distances. Only pairwise distances between samples from the second sequencing batch, the same country, the same season and different participants were included in **c** and **d** (650 RSV-A pairs and 656 RSV-B pairs). Each dot represents an individual sample in **a** and **b**, and a sample pair in **c** and **d**. Two-tailed Mann–Whitney U-tests were used to evaluate the significance of the differences. P values are shown above the plots. For **a** and **b**, the centre line of each box denotes the median; box limits, the first and third quartiles; whiskers, the highest and lowest values within 1.5 times the interquartile range from the box limits and outlying points, outliers. For **c** and **d**, the violin plots summarise the distribution of the data, and the black dots denote the median value of each group.

NS2, P, SH and G genes, but the other genes did not have significant differences in pairwise nucleotide diversity between each other (Supplementary Fig. 4). These significant differences were by definition due to the mean proportion of pairwise nucleotide differences at each genomic position within the L gene instead of the length of the L gene.

RSV-B had greater pairwise nucleotide diversity than RSV-A after adjusting for the batch effects (multiple linear regression, $P = 0.044$, Supplementary Table 2 and Fig. 2a), and older adults had a more diverse intrahost RSV-B population than infants (multiple linear regression, $P = 0.0006$, Supplementary Table 2 and Fig. 2b). The subgroup difference was still significant if excluding adult samples (Mann–Whitney U-tests on standardised data, $P = 0.039$). The number of RSV reads and the duration between symptom onset and sample collection were similar between both RSV subgroups and between both age groups. Samples collected from different countries or seasons or patients with different severity of RSV infections did not have significant differences in pairwise nucleotide diversity (Supplementary Table 2).

**Genetic distance**. Within-host diversity levels between samples were compared using pairwise Manhattan distances[24] at consensus-identical positions, where allele frequencies below the 3% threshold were converted to 0. In contrast, consensus variations between samples were compared using pairwise patristic distances, which are phylogenetic distances on RSV phylogenies (Supplementary Fig. 5). To eliminate the batch effects, we only included pairwise distances between samples in the second batch ($n = 112$; excluding one outlier). To reduce potential bias from geographical and temporal differences, only pairwise distances between samples from the same country and the same season were calculated.

Serial sample pairs (i.e. pairs with both samples collected from the same participant) had within-host diversity levels comparable to those of samples from different participants (range: 0–3.34 vs. 0–5.03), despite having identical or nearly identical consensus sequences, as indicated by their small patristic distances (range $2.0 \times 10^{-6} - 7.5 \times 10^{-5}$). Excluding the serial sample pairs, RSV-B sample pairs had significantly greater within-host diversity levels than RSV-A pairs (median: 1.24 vs. 0.86), whereas the comparison of consensus sequences showed the opposite effect (Fig. 2c, d). Pairwise patristic distances between RSV-A samples

formed three clusters, corresponding to the three main clades of the phylogenetic tree (Supplementary Fig. 5a). When using all allele frequencies, including those below 3% MAF, to calculate Manhattan distances, RSV-B sample pairs still had significantly greater pairwise Manhattan distances than RSV-A pairs (median: 20.5 vs. 18.2, $P = 8.2 \times 10^{-58}$; Supplementary Fig. 6).

**Temporal change of intrahost virus population**. Putting all samples together, standardised pairwise nucleotide diversity did not have a significant temporal change within 7 days of symptom onset ($R^2 = 0.008$; $P = 0.122$). For the 34 participants with multiple samples collected daily during hospitalisation, pairwise nucleotide diversity was also evaluated in each set of serially collected samples, excluding those sequenced in different batches (Fig. 3). No significant trend was noted either in each participant or when combining all samples and adjusting for the batch effects. The only exception was the samples from GB-058, where pairwise nucleotide diversity increased by 0.000063 daily (95% confidence interval, 0.000046 to 0.000080; $P = 0.004$). This patient was a 19-day-old preterm neonate (gestational age of 33 weeks 6 days) with severe RSV infection requiring intensive care and mechanical ventilation.

The changes in minor variants and variant frequencies in the serial samples were also evaluated at polymorphic sites where minor alleles were identified at more than three time points (Fig. 4). Of these minor variants, 79% had a nonsynonymous substitution. Only one minor variant with a G to A substitution at position 3403 of the L gene from participant NL-091, which was shared by 71 participants (85 samples), remained above the 3% threshold throughout the sampling period. This patient was a 42-day-old previously healthy infant with severe RSV infection requiring intensive care and mechanical ventilation. All other variants (including the aforementioned variant in other participants) were only detected either early, late or intermittently during the course of sample collection.

**Discussion**

In this study, we sequenced 858 nasopharyngeal samples collected in three clinical studies during 2017–2020 and profiled within-host RSV populations from 319 samples. We demonstrated that

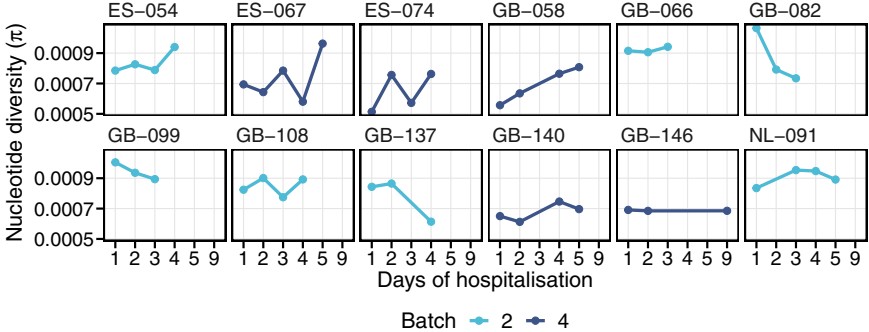

**Fig. 3 Temporal change of pairwise nucleotide diversity.** Pairwise nucleotide diversity of serial samples collected at more than two time points and sequenced in the same batch are shown here. Three participants whose samples were sequenced in different batches and 19 participants who had only two samples collected are not shown. Each panel is labelled with the participant ID.

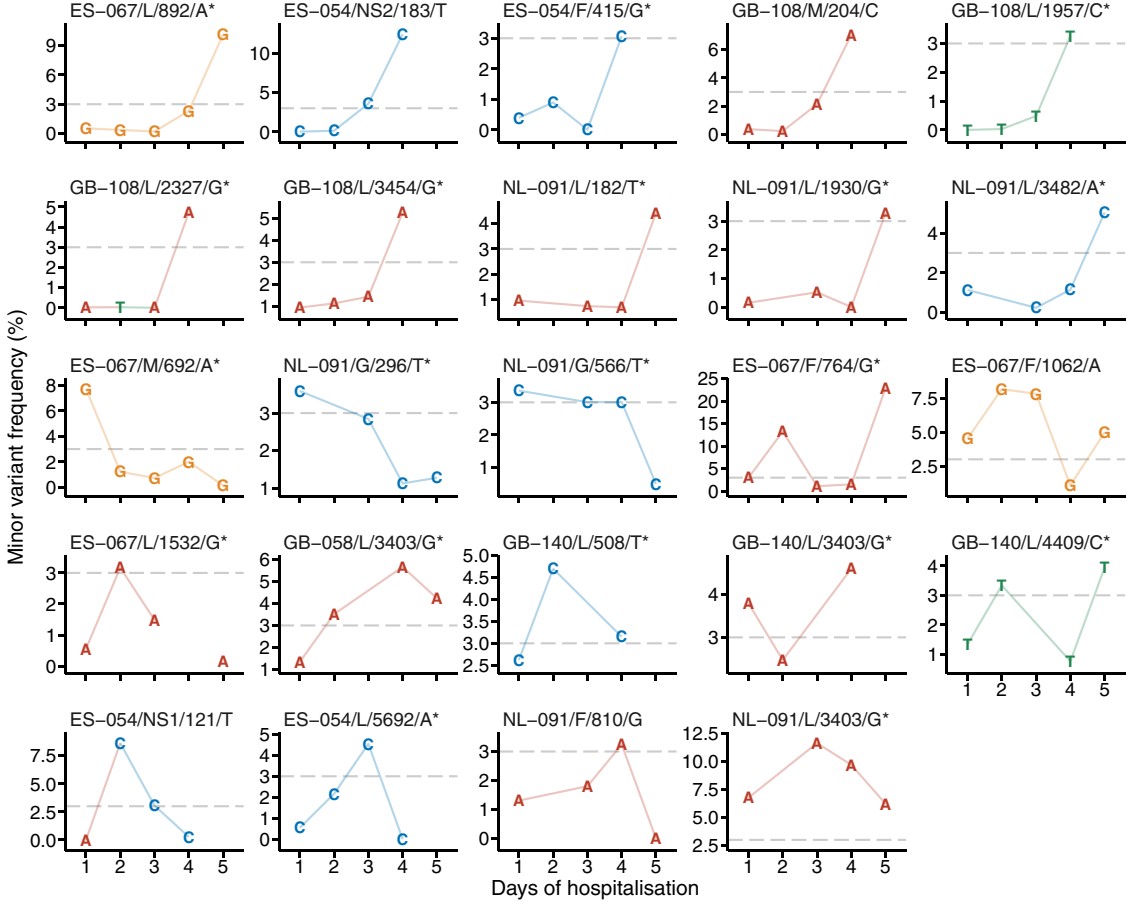

**Fig. 4 Temporal change in minor alleles.** Minor alleles and allele frequencies are shown at polymorphic sites within the coding sequence of the serial samples, where minor alleles were detected at ≥3 time points. The grey dashed lines represent the 3% threshold, which defines a minor variant. Panels are labelled with the participant ID, followed by the gene name, the nucleotide position and the consensus base. Asterisks denote nonsynonymous substitutions. Letters in the plots denote minor allele bases. Panels are ordered by the trend of the change: increased, decreased and fluctuated. Positions are numbered from the first base of the coding sequence of each gene according to the NCBI reference sequences with the accession numbers of NC_038235 and NC_001781 for RSV-A and RSV-B, respectively.

RSV-B had greater within-host diversity than RSV-A, whereas RSV-A had greater consensus diversity than RSV-B. Two RSV-B isolates' consensus sequences had a mutation in the F protein (K68N), previously associated with reduced susceptibility to nirsevimab neutralisation. Several other minor variants were also identified in the antigenic sites of the F protein. None of these variants have been reported before except for S255N[25], whose susceptibility to monoclonal antibodies has not been examined.

Stochastic (random) patterns were found in the temporal changes of within-host virus diversity and minor variants.

Low input genetic material (i.e. viral load) has been shown to reduce the sensitivity and specificity of variant calling[26]. In this study, we applied the quantitative methodology of targeted metagenomics to library construction and used the number of unique RSV reads as a proxy for viral load[27]. The inclusion criterion of more than 10,000 unique RSV reads corresponded with

a viral load of ~$2.4 \times 10^6$ copies/mL and above, sufficient input levels for accurate minority variant calling[28]. Given a large number of samples in this study, batching was required for sequencing, resulting in variable percent duplication rates and hence some batch effects on diversity metrics. We adopted two approaches to account for the batch effects on the comparisons of mean cumulative MAF per sample and pairwise nucleotide diversity: (i) including batch as a regression covariate and (ii) standardising the values within each batch to $z$-scores (see Methods for details). Both methods showed the same significant findings, making cross-batch comparisons robust. To avoid any residual bias, for pairwise comparisons of genetic distances we used only samples from the same batch (batch 2), which had very high percent duplication rates and similar read counts for RSV-A and RSV-B (Table 1 and Supplementary Table 1), consistent with capture saturation, and from which we could be confident of recovering the full range of intrahost diversity.

The extent of intrahost virus diversity depends not only on the rate of virus evolution (partly associated with the ability of proofreading for viral replication errors) but also on the duration of infection. RNA viruses generally have a higher mutation rate than DNA viruses[29], and are usually not able to correct the errors of viral replication, which DNA viruses can[30]. In our study, RSV had greater pairwise nucleotide diversity than has been reported for influenza virus, another RNA virus causing acute respiratory infection (range 0.0005–0.0014 vs. 0–0.0002[31]). RSV intrahost diversity appears to be comparable with, or slightly higher than, that of the DNA viruses in the family *Herpesviridae*, which cause chronic infections[32], but up to one to two orders of magnitude lower than that of persistent RNA viruses (e.g. hepatitis C virus and human immunodeficiency virus) and persistent DNA viruses (e.g. hepatitis B virus), which generally have pairwise nucleotide diversity above 0.005[32].

Neutralisation escape mutants have been isolated in 0.7% of immunoprophylaxis-naïve RSV-infected subjects[13], 5–9% of RSV-breakthrough patients receiving palivizumab[12,33] and 8% of RSV-breakthrough cases receiving nirsevimab[34]. In our study, isolates collected from 0.8% (2/264) of the immunoprophylaxis-naïve participants were found to contain a nirsevimab resistance-associated substitution at the consensus level. We also identified an RSV-B minor variant with an amino acid change from serine to proline at position 275 (S275P) of the F protein. Other amino acid substitutions at this position have demonstrated resistance to palivizumab (S275F/L)[12]. Whether the mutation S275P also alters the neutralising activity of palivizumab requires further investigation; however, all three mutations at this position replaced a polar amino acid with a nonpolar one, which may result in significant conformational or functional changes. It is important to identify neutralisation escape mutants in immunoprophylaxis-naïve children in the era before RSV monoclonal antibodies become extensively used. It indicates the circulation of escape mutants in the community even though they generally have a selective disadvantage in the absence of monoclonal antibodies[13].

Our findings that RSV-B had greater pairwise nucleotide diversity and pairwise Manhattan distances than RSV-A both indicate that, at least in our dataset, RSV-B had a more diverse intrahost virus population than RSV-A. These results do not correlate with the duration between symptom onset and sample collection (Table 1), but are consistent with previous studies on global RSV strains, which found that RSV-B has a higher genome-wide evolutionary rate than RSV-A ($7.47$–$7.76 \times 10^{-4}$ substitutions/site/year vs. $5.68$–$6.47 \times 10^{-4}$ substitutions/site/year)[35,36]. This difference extends below the 3% threshold for minority variant calling (Supplementary Fig. 6). On the basis of these findings, we hypothesise that RSV-B is subject to greater immune pressure (e.g. by innate immunity, neutralising antibodies or T cell-mediated cytotoxicity) than RSV-A. This hypothesis is in line with previous studies showing that intrahost RSV diversity increased in response to an established immunity[21] and that RSV-B has more amino acid alterations[37], predicted O glycosylation site changes[37] and indel mutations[36] in the G gene than RSV-A, suggesting a stronger selective pressure acting on RSV-B than on RSV-A.

RSV-B exhibited higher within-host diversity in older adults than in infants in response to different immune pressures between the two age groups. Of note, our dataset included only eight adults, and this comparison was limited to seven adult samples and 137 infant samples collected from those with RSV-B infection. Further studies enrolling more adults would be of value to delineate the difference in within-host diversity between different age groups. Furthermore, the temporal changes of pairwise nucleotide diversity and minor variants were stochastic within each infected individual, suggesting the driving force of evolutionary dynamics in global RSV populations is more likely from the selective pressure imposed at the population level than within an individual host. Only samples that yielded sufficient RSV reads were included in this study, so these temporal trends were confined to samples collected over a short time frame (mostly within 5 days of symptom onset). Nonetheless, a study on seasonal influenza virus also found limited evidence of positive selection at the within-host evolutionary scale[24].

The greater within-host virus diversity observed in RSV-B than in RSV-A warrants separate testing and close monitoring of the anti-RSV-B efficacy of vaccines and monoclonal antibodies that are being developed. This is because the development of several RSV vaccines in preclinical or clinical trials is based on the nucleotide sequences or structure of RSV-A strains[38–40]. Some studies have also shown that RSV-B had more fixed mutations in the antigenic sites of the F protein at the consensus level[41], resulting in more variable in vitro and clinical susceptibility to monoclonal antibodies than RSV-A. For example, in a phase 2b trial of nirsevimab, the drug had reduced neutralising activity against two RSV-B isolates collected from its recipients; one had a mutation of N208S and the other had multiple mutations of I64T, K68E, I206M and Q209R in the F protein[34]. A phase 3 trial of another investigational RSV monoclonal antibody, suptavumab, failed to meet its primary end point because all RSV-B strains identified in the trial carried two amino acid changes in the F protein (L172Q and S173L), conferring resistance to the drug[8]. All RSV-B samples in our study also harboured these two amino acid substitutions, except for one that encoded isoleucine instead of leucine at position 173 (a nonpolar-to-nonpolar substitution).

We excluded genomic positions where consensus bases were different from the calculation of Manhattan distance, to ensure that between-host genetic distance would be driven by differences in minor alleles rather than differences at the consensus level[24]. We found that, outside the consensus-different positions, serial samples from the same individual did not have a shorter pairwise Manhattan distance than that of a randomly taken between-host pair from the same country and season. This methodology change makes our results robust to inter-host variation, in contrast to previous studies on influenza virus and RSV, where distance metrics were largely driven by consensus differences[42,43].

Our findings suggest that RSV-B has a more diverse within-host population than RSV-A, likely driven by selection pressure at the host-population level. This difference between the two subgroups warrants close monitoring of vaccine efficacy and emergence of neutralisation escape variants.

## Methods

**Sample collection**. Nasopharyngeal swabs were collected from patients with respiratory symptoms under 1 year old or over 60 years old, from London and

Oxford, United Kingdom, Santiago de Compostela, Spain and Utrecht, the Netherlands, during 2017–2020. These patients were enrolled in three clinical studies of the REspiratory Syncytial virus Consortium in EUrope project (RESCEU, ClinicalTrials.gov identifiers: NCT03627572[44], NCT03756766[45] and NCT03621930[46]), a European multicentre project investigating epidemiological, virological and immunological characteristics of RSV infection. None of these participants had received any RSV monoclonal antibody or investigational vaccine. RSV infection was diagnosed using molecular point-of-care testing on the Alere™ i RSV platform (Abbott, Illinois, US) in infant participants and on the GeneXpert® influenza/RSV system (Cepheid, California, US) in adult participants in a community setting, and using antigen and/or PCR tests at a central laboratory in a hospital setting. A nasopharyngeal swab was collected from each participant within 7 days of symptom onset, and daily swabs were also collected from RSV-positive hospitalised infant participants where possible until hospital discharge. After collection, swabs were immersed in an M4RT® transport medium, aliquoted, and frozen at −80 °C until use.

The severity of an RSV infection was defined using the ReSVinet scale[47] in infants. This scale accounts for several clinical variables, including feeding intolerance, medical intervention, respiratory difficulty, respiratory frequency, apnoea, general condition and fever. The score ranges from 0 to 20; a score of 0–7 was defined as mild, a score of 8–13 as moderate and a score of 14–20 as severe. In older adults, those who did not require any treatment or medical attendance were defined as having mild disease, those requiring hospitalisation were defined as having severe disease and the rest were defined as having a moderate RSV disease.

These clinical studies were conducted in accordance with the provisions of the Declaration of Helsinki and were approved by the relevant ethics committees at each site, including the University of Oxford, the Health Research Authority (IRAS IDs: 224156 and 231136), the NHS National Research Ethics Service Oxfordshire Committee A (reference number: 15/SC/0335), the South Central and Hampshire A Research Ethics Committee (reference number: 17/SC/0522) and the London-Central Research Ethics Committee (reference number: 17/LO/1210) in the UK; Hospital Clínico Universitario of Santiago de Compostela, and Comité de Ética de la Investigación de Santiago-Lugo (reference number: 2017/395) in Spain; the Medical Ethical Committee, University Medical Center Utrecht (reference number: 17/563) and the Ethical Review Authority (reference number: NL60910.041.17) in the Netherlands. All adult participants and the parents or guardians of all infant participants provided written informed consent.

**Nucleic acid isolation and whole-genome sequencing.** All RSV-positive samples were selected for whole-genome sequencing. Nucleic acid isolation, library construction and sequencing were performed in four different batches. To minimise the risk of RNA degradation, nucleic acid was extracted locally from primary samples, and the extractions were scheduled as close as practical to the time of sequencing.

Total nucleic acid extraction was carried out using the NucliSENS® easyMAG® system (BioMérieux, Marcy-l'Étoile, France), following the manufacturer's instructions. Wherever possible, 500 μL of each sample was used to get 25 μL eluate in the first and fourth batches, and 35 μL in the second and third batches.

Sequencing libraries were constructed using the methodology of targeted metagenomics[27], a modification of the veSEQ-HIV protocol[48]. A 12-μL aliquot of each nucleic acid sample was first concentrated to 3 μL with RNAClean XP magnetic beads (Beckman Coulter, California, United States). Dual-indexed libraries for Illumina sequencing were then constructed using the SMARTer Stranded Total RNA-Seq Kit v2 - Pico Input Mammalian (Takara Bio USA, California, United States), where first-strand reverse transcription was primed with tagged random hexamers and double-stranded cDNA was synthesised with sets of i5 and i7 index primers, as previously described elsewhere[49]. These gave unique dual indexing (UDI) for the samples, thus minimising the risk of index misassignment during sequencing. After 12 cycles of PCR amplification of the cDNA, 10 μL of each library was pooled and purified using AMPure XP (Beckman Coulter). A 750-ng aliquot was taken from the pool and captured using a predesigned SureSelect RNA Target Enrichment multi-pathogen probe set (Agilent, California, United States). This probe set (each 120 nucleotides long) targeted more than 100 pathogenic bacteria and viruses, including both RSV-A and RSV-B[50]. Sixteen cycles of PCR were performed for post-capture amplification, and the final product was purified by AMPure XP.

Sequencing was performed on the Illumina MiSeq platform (Illumina, California, US) with the MiSeq Reagent Kit v3 (600-cycle) for the first and third batches, generating 265-bp and 300-bp paired-end reads, respectively. The second and fourth batches were sequenced on the Illumina NovaSeq 6000 system with the NovaSeq 6000 SP Reagent Kit v1.5 (300-cycle), generating 151-bp paired-end reads.

**Genome reconstruction.** The first six bases of read 1 and the first three bases of read 2 were clipped off to remove random hexamer primers and the SMARTer adaptor sequences, respectively. An extra three bases at the 5′ end of MiSeq-generated read 2 were also cut off as they had reduced quality. Trimmomatic (v0.39)[51] was then used to trimmed off adaptor sequences and low-quality bases with a Phred score below 20 (option: Adaptors:2:10:7:1:true LEADING:20 TRAILING:20 SLIDINGWINDOW:4:20 MINLEN:50). De novo assembly of the

trimmed reads was carried out using both IVA (v1.0.8)[52] and SPAdes (v3.14.1)[53], in each case selecting the contig sequences with a higher N50 for genome reconstruction using shiver[54]. Internally, BLASTN (v2.7.1+)[55] was used for read and contig classification, MAFFT (v7.471)[56] was used for sequence alignment and Bowtie 2 (v2.4.1)[57] was used for read alignment (option: --very-sensitive-local). A minimum base quality of 35 and mapping quality of 30 were required for a base or an alignment to be counted as mapped. Mapped RSV reads were deduplicated with Picard MarkDuplicates (v2.18.14, https://broadinstitute.github.io/picard/). Pre-deduplicated per-position mapped read counts, generated by shiver, were used for downstream within-host virus diversity analysis.

**Within-host virus diversity analysis.** Only samples generating more than 10,000 unique (i.e. deduplicated) RSV reads and containing a single subgroup of RSV were included in within-host virus genetic diversity analysis. We have previously shown that RSV viral load highly correlates with the number of unique RSV reads generated by this sequencing method[27], consistent with high-quality RNA being recovered in a quantitative way. Ten thousand unique RSV reads correspond to a viral load of ~2.4 × 10^6 copies/mL. Allele frequencies were calculated at each genomic position, excluding those supported by fewer than 200 reads. The choice of this cut-off was based on a predefined criterion that 90% of the included samples had at least 80% of the genome fulfilling this cut-off (Supplementary Fig. 7). Cumulative MAF was defined as 1 minus major allele frequency, and polymorphic sites were those with a cumulative MAF of ≥3%. Mean cumulative MAF per sample was calculated as the sum of cumulative MAF at each genomic position divided by the total number of positions. Minor variants, or intrahost single nucleotide variants, were defined as variants with an allele frequency of ≥3% and <50%.

Intrahost virus diversity was estimated as pairwise nucleotide diversity ($\pi$)[58]. The proportion of pairwise nucleotide differences ($D$) at each genomic position was calculated as

$$D_i = \frac{A_i \times C_i + A_i \times G_i + A_i \times T_i + C_i \times G_i + C_i \times T_i + G_i \times T_i}{(N_i^2 - N_i)/2} \quad (1)$$

where $A_i$, $C_i$, $G_i$ and $T_i$ represent the copy number of allele A, C, G and T, respectively, and $N_i$ is the total count of the four alleles (i.e. depth of coverage) at a given locus $i$, so $N_i = A_i + C_i + G_i + T_i$. Loci with a total count of less than 200 were excluded. Pairwise nucleotide diversity across a genome ($\pi$) was then calculated as

$$\pi = \sum_{i=1}^{L} \frac{D_i}{L} \quad (2)$$

where L is the number of genomic positions with a read depth of at least 200×.

Manhattan (L1-norm) distance was used to compare within-host diversity levels between samples, calculated as

$$d_i(\mathbf{p}, \mathbf{q}) = \sum_{k=1}^{4} |\mathbf{p}_k - \mathbf{q}_k| \quad (3)$$

$$M = \sum_{i=1}^{N} d_i \times \frac{S}{N} \quad (4)$$

where $d_i$ is the distance between two samples at a given locus $i$ with vectors $\mathbf{p}$ and $\mathbf{q}$ containing relative frequencies of four possible alleles (i.e. A, C, G and T), $M$ is the Manhattan distance between the coding sequences of two samples, $N$ is the number of coding sequence positions where both samples have the same consensus base and a read depth of at least 200× and $S$ is the total length of the coding sequence. To remove potential background noise in Manhattan distance calculations, allele frequencies of <3% were changed to 0, and those of >97% were changed to 100%.

Nucleotide positions were numbered from the first base of the coding sequence of each gene according to the NCBI reference sequences with the accession numbers of NC_038235 and NC_001781 for RSV-A and RSV-B, respectively. Amino acid positions were numbered from the first methionine of each protein according to the same NCBI reference sequences.

**Phylogeny reconstruction.** Maximum likelihood phylogenies of consensus coding sequences, supported by at least two unique (deduplicated) RSV reads, were estimated using RAxML (v8.2.12)[59] with the general time-reversible nucleotide substitution model and gamma-distributed rate heterogeneity. Bootstrapping with 1000 replicates was used to assess the robustness of tree topologies. Pairwise patristic distances were calculated from the maximum-likelihood trees using the cophenetic function of the R package ape (v5.4-1)[60]. Phylogenetic trees were visualised using the R package ggtree (v2.2.4)[61].

**Statistical analysis.** Continuous variables were summarised using mean, median, maximum and minimum. All comparisons of continuous variables between groups were conducted by two-tailed Mann–Whitney U-tests (two groups) or Kruskal–Wallis tests (three groups). Post hoc application of the Benjamini–Hochberg procedure was used to control false discovery rates for multiple testing. Chi-square tests with Yates' continuity correction were used for contingency analysis; Fisher's exact tests were performed when the expected value of a cell was less than 5. Logistic regression was employed to model a binary

dependent variable while adjusting for a covariate. Two-tailed Pearson correlation analysis was used to evaluate the relationship between two variables. Temporal changes of a variable were determined by ordinary least-squares linear regression. Two approaches were applied to account for batch effects on the comparisons of diversity metrics: (i) including batch as a regression covariate (e.g. regression of pairwise nucleotide diversity on sampling country, sampling season, RSV subgroup, RSV read count, participant age group, disease severity and 'batch' as in Supplementary Table 2); and (ii) standardising the values within each batch to $z$-scores, that is, to a mean of zero and a standard deviation of 1 (e.g. Mann–Whitney $U$-test on $z$-score standardised pairwise nucleotide diversity as in Fig. 2). Missing data were imputed using the aregImpute function, implemented in the R package Hmisc (v4.5-0)[62]. All statistical analyses were performed using R (v4.0.2)[63]. $P$ values or adjusted $P$ values of less than 0.05 were considered to indicate statistical significance.

**Reporting Summary**. Further information on research design is available in the Nature Research Reporting Summary linked to this article.

## Data availability

The sequencing read data generated in this study have been deposited in the European Nucleotide Archive under study accession PRJEB34042. The RSV genomic sequences generated in this study have been deposited in GenBank under accession numbers LR699315 LR699726, LR699734, LR699736-LR699744 and MZ515551-MZ516143. The RSV reference sequences used in this study are available in GenBank under accession numbers NC_038235 and NC_001781. The associated sample and de-identified clinical information used in this study is provided in Supplementary Data 1.

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

## Acknowledgements

This work was supported by the National Institute for Health Research (NIHR) Oxford Biomedical Research Centre, the NIHR Thames Valley and South Midlands Clinical Research Network, the British Research Council, and the REspiratory Syncytial virus Consortium in EUrope (RESCEU) project. RESCEU has received funding from the Innovative Medicines Initiative 2 Joint Undertaking (grant number 116019). This Joint Undertaking receives support from the European Union Horizon 2020 Research and Innovation Programme and the European Federation of Pharmaceutical Industries and Associations.

## Author contributions

G.-L.L., T.G. and A.J.P. conceived and designed the work. G.-L.L., S.B.D., M.D.S., D.Ö., J.A., C.B., L.B., P.O., F.M.-T., H.N. and A.J.P. conducted and supervised the clinical studies. M.A.A. designed the probe set that was used for capture. M.d.C., D.B. and R.B. designed the sequencing protocol. G.-L.L., A.B., G.M.-C., E.M.-G. and M.d.C. performed the experiments. G.-L.L., T.G., D.O'C. and A.J. analysed and interpreted the data. G.-L.L. drafted the manuscript and T.G., D.O'C. and A.J.P. substantively revised it. T.G. and A.J.P. supervised the work. All authors have approved the submitted version and agreed to submit the manuscript.

## Competing interests

S.B.D. has been an investigator for clinical trials of vaccines and antimicrobials for pharmaceutical companies including AstraZeneca, Merck and Janssen, and sits on an RSV advisory board for Sanofi Pastuer. M.A.A. is supported by a Sir Henry Dale Fellowship jointly funded by the Royal Society and Wellcome Trust (220171/Z/20/Z). D.Ö. and J.A. are employees of Janssen Pharmaceutica NV. F.M.-T. has received honoraria from GSK, Pfizer Inc., Sanofi Pasteur, MSD, Seqirus and Janssen for taking part in advisory boards and expert meetings and for acting as a speaker in congresses outside the scope of the submitted work. F.M.-T. has also acted as principal investigator in randomised controlled trials of the above-mentioned companies as well as Ablynx, Regeneron, Roche, Abbott, Novavax and MedImmune, with honoraria paid to his institution. F.M.-T. receives support for his research activities from the Instituto de Salud Carlos III (Proyecto de Investigación en Salud, Acción Estratégica en Salud): Fondo de Investigación Sanitaria (FIS;PI1601569/PI1901090) del plan nacional de I+D+I and 'fondos FEDER'. A.J.P. is a National Institute for Health Research (NIHR) Senior Investigator with funding from the British Research Council. The remaining authors declare no competing interests. The views expressed in this article are those of the authors and may not be understood or quoted as being made on behalf of or reflecting the position of the organisations with which the authors are employed/affiliated.

## Additional information

## RESCEU Investigators

Harry Campbell[13], Steve Cunningham[13], Debby Bogaert[8,14], Philippe Beutels[15], Joanne Wildenbeest[8], Elizabeth Clutterbuck[1], Joseph McGinley[1], Ryan Thwaites[10], Dexter Wiseman[10], Alberto Gómez-Carballa[12],

Carmen Rodriguez-Tenreiro[12], Irene Rivero-Calle[12], Ana Dacosta-Urbieta[12], Terho Heikkinen[16], Adam Meijer[17], Thea Kølsen Fischer[18], Maarten van den Berge[19], Carlo Giaquinto[20], Michael Abram[21], Philip Dormitzer[22], Sonia Stoszek[23], Scott Gallichan[24], Brian Rosen[25], Eva Molero[26], Nuria Machin[26] & Martina Spadetto[26]

[14]Queen's Medical Research Institute, University of Edinburgh, Edinburgh, UK. [15]Centre for Health Economics Research and Modelling Infectious Diseases, Vaccine and Infectious Disease Institute, University of Antwerp, Antwerp, Belgium. [16]Department of Pediatrics, University of Turku, Turku University Hospital, Turku, Finland. [17]National Institute for Public Health and the Environment, Bilthoven, Netherlands. [18]Statens Serum Institut, Copenhagen, Denmark. [19]Department of Pulmonary Diseases, University of Groningen, University Medical Center Groningen, Groningen, Netherlands. [20]PENTA Foundation, Padua, Italy. [21]AstraZeneca, Gaithersburg, MD, US. [22]Pfizer, Pearl River, NY, US. [23]GlaxoSmithKline, Potomac, MD, US. [24]Sanofi Pasteur, Toronto, Ontario, Canada. [25]Novavax, Potomac, MD, US. [26]Team-It Research, Barcelona, Spain.

