## [Peer Review File · Nature Communications]

REVIEWER COMMENTS

Reviewer #1 (Remarks to the Author):

The manuscript by Lin et al. Presents novel findings regarding the in-host genomic diversity of RSV-A and RSV-B. The manuscript is well written, engaging and contains a comprehensive description of the methodology and analysis used. Overall this is a high quality manuscript describing novel findings and would be of high interest to the virology and clinical fields.

The major findings of the manuscript are the lower consensus level diversity of RSV-B compared to RSV-A. However RSV-B exhibits higher within-host diversity than RSV-A. Importantly some of the sub-consensus and consensus diversity occurs in antigenic sites and may confer resistance to licensed and pre-clinical monoclonal antibodies. Understanding population and within-host diversity is important to ensure the effectiveness of future RSV vaccines and therapeutics.

I thank the authors for their detailed and comprehensive description of the methods and results.

The sample cohort was reduced from 861 NPA swabs to 319 in the final data set. The authors attribute this loss to quality limits on the number of de-duplicated reads required for high confidence MAF calling. However, was the viral load investigated? Did low quality of the original RNA extracted also contribute?

Given the inverse correlation between the number of minority variants and the number of unique reads (Figure S2). What is the justification of establishing a read depth cut off of 200x to call minority variants at any base position?

Did you describe your upper limit for MAF? In Figure 1. The x-axis indicated that MAF were called from >3% - <50% is this correct? Please add this detail.

Limitations when comparing the RSV genome diversity between adults and children should be included considering the cohort consists of 258 children and 9 adults. For example, on page 7 line 211 and page 9 line 267.

The temporal within-host diversity is very interesting, although the authors do not describe some of the co-founders here. Median sampling of virus occurred 4 days after symptom onset and at most continued to 8 days post symptom onset. Therefore sampling over the course of any RSV infection was limited and within host diversity maybe occurring as the infection is established within the host. I recognize the difficulties here in terms of viral detection and establishing sufficient viral load to produce sequencing read depth. However I feel this limitation should be acknowledged.

Reviewer #2 (Remarks to the Author):

Respiratory syncytial virus (RSV) is a major cause of disease in children and the elderly. There are two main subtypes: A and B, and many different genotypes and strains. There have been studies regarding the evolution and diversity of RSV within a population, but little is known about the diversity of an RSV population within each host. Lin, et al, examine the question of RSV genetic diversity within a host. To accomplish this, the group deep sequenced the RSV population from nasopharyngeal samples collected from participants in three clinical studies of the RESCEU project. The authors identified significant differences based on patient age and virus subgroup.

The authors have a clear objectives: to study the RSV population at the host level. It has plainly described findings: that the RSV-B subgroup exhibited more diversity within-host, while RSV-A exhibited more diversity at a consensus level. The authors paid particular attention to the F gene, which codes for the fusion protein – the major target for neutralizing antibodies. The authors found the F gene to be largely genetically stable at the consensus level, much as other groups have. However, there were minor variants present in the antigenic sites, and one mutation found was in two participants had been shown in a previous study to cause a significant reduction in susceptibility to nirsevimab. Any genetic changes in this gene may have wide-ranging implications in vaccine, monoclonal antibody, and targeted small molecule development and suggests the need for continued genetic surveillance on RSV infections (particularly RSV-B) to monitor minor variants.

The authors could have made the paper more broadly accessible by explaining a bit more. For example, on page 8, line 224-227, two methods for controlling for batch effects are described but not in a way that I could understand. And a cursory search online did not yield an answer. An explanation that includes "...per sample and nucleotide diversity: (i) including batch as a regression covariate of [what] and [what]" would help. Another example is that it would strengthen the argument to mention that pairwise Manhattan distance calculation was used as a test for intrahost diversity and the pairwise patristic calculation was testing for consensus variation.

The authors used targeted metagenomics and used the number of unique RSV reads as a proxy for viral load to control for any variation caused by lower viral loads/unique reads. However, this method was described for influenza. Because of the importance of this control, it would be worthwhile to show that these factors do indeed correlate with viral load so that it can be confidently used as a proxy.

The authors found that genetic diversity of RSV in older adults was higher than in infants (Fig 2b). However, can this truly be determined when the adult patient cohort was so much smaller than the infant cohort? It is an interesting finding and could perhaps lead to a new study, but I believe it is premature to conclude anything based on so few data points.

The study found that the difference in intrahost diversity between RSV-A and RSV-B is not correlated with the duration between symptom onset and sample collection. It would be interesting to know if there is a correlation with clinical severity of the infection.

On page 9, line 265-267, the authors suggest that RSV-B is subject to greater immune pressure than RSV-A. This is an intriguing possibility that could be expounded upon.
Specific Issues:

There is no delineation between the Abstract and the Introduction.

p.2, l.51-52. the line sounds awkward and would flow better if it read, "...there is no efficacious antiviral for treatment or licensed vaccine to prevent RSV infection..."

p.2, l.63. here, and in other places, you write "antigenic sites (neutralizing epitopes)", which makes it sound like antigenic sites are being equated to neutralizing epitopes, rather than what I believe you are saying, which is that you are speaking of neutralizing epitopes in particular.

p.3, l.118. it is unclear what the duplication rate refers to. Is it the sequential samples from the same patient? Or is it duplicate reads?

l.143. The first sentence in this paragraph makes no sense until the reader gets to the end and then

needs to reread it to understand it.

I.146. One minor variant equals 0% minor variants?

I.147. "Two of these minor covariants coexist in the same sample...substitution from Ile to Thr." Are the two covariants in one codon that leads to the AA change, or distant from each other and only one of them causes an AA change?

I.154. Are "pairwise nucleotides" neighboring nucleotides or anywhere in the genome?

I.159. "The L gene had significantly higher nucleotide diversity than..." Is that because the L gene is the largest gene? Or is this based on frequency within the gene?

Supplementary figures 1, 2, and 3 describe the results in the figure legend. They should only describe the figure if the results are described in the body.

Reviewer #3 (Remarks to the Author):

This study examines intra-host diversity of human respiratory syncytial virus (RSV). Using deep-sequencing of nasopharyngeal swabs from 3 countries during consecutive RSV seasons, the authors find that intra-host diversity differs between the two RSV subgroups, inversely from their population level diversity. Longitudinal samples of a subset of the infected infants showed mostly nonlinear changes over time in diversity.

The data are interesting and are well-presented. Intra-host diversity of acute RNA viral infections in general remains an understudied field. This manuscript will enhance our understanding of these infections.

There are a few minor concerns whose clarification would enhance the manuscript.

1. Figure 1. Analysis of G diversity provided as a counterpoint to F would be beneficial as G is also an antigen and has previously been reported to have hypermutation at the population level in the C-terminus.
2. Figure 2. It is not clear how much the analysis of older adults contributes to the impact of the overall data set given the large disparity in sample sizes between that population and infants. A focus on infant samples would mitigate the confounding factor of prior immunity.
3. Figure 3. The text states that 34 participants had multiple samples collected daily but only 12 individuals are shown. If the samples for 22 participants were excluded from the analysis due to sequencing in different batches, that would be important to note in the text.
4. Figure S2: There appear to be many more minor variants in Batch 2 compared to the other batches, even adjusting for overall number of unique reads.

The reviewers' comments were extremely helpful and we have used them to improve the manuscript. We apologise for the delay in resubmitting the manuscript since we opted to carry out additional qPCR assays to demonstrate the correlation between deduplicated RSV read counts and viral load, which we have included in the response. We have replied to the comments point-by-point as follows (the line numbers are indicated according to the 'track changes' version of the manuscript):

Response to Reviewer #1:

The manuscript by Lin et al. presents novel findings regarding the in-host genomic diversity of RSV-A and RSV-B. The manuscript is well written, engaging and contains a comprehensive description of the methodology and analysis used. Overall this is a high quality manuscript describing novel findings and would be of high interest to the virology and clinical fields.

The major findings of the manuscript are the lower consensus level diversity of RSV-B compared to RSV-A. However RSV-B exhibits higher within-host diversity than RSV-A. Importantly some of the sub-consensus and consensus diversity occurs in antigenic sites and may confer resistance to licensed and pre-clinical monoclonal antibodies. Understanding population and within-host diversity is important to ensure the effectiveness of future RSV vaccines and therapeutics.

I thank the authors for their detailed and comprehensive description of the methods and results.

We thank the reviewer for the positive assessment and are grateful for the constructive feedback.

The sample cohort was reduced from 861 NPA swabs to 319 in the final data set. The authors attribute this loss to quality limits on the number of de-duplicated reads required for high confidence MAF calling. However, was the viral load investigated? Did low quality of the original RNA extracted also contribute?

Low RNA integrity can indeed reduce read numbers, although as a capture-based protocol with short (120nt) probes, veSEQ-Castanet (the methodology we used for target enrichment) is relatively robust to RNA degradation. In our study, we minimised the risk of RNA degradation by extracting RNA locally from primary samples, and timing extractions as close as practical to the time of sequencing. We also performed RSV RT-qPCR to quantify viral load on the same RNA extracts used for sequencing in our previously published work.¹ We showed that RSV viral load highly correlates with the number of deduplicated RSV reads, as has been previously reported for other applications of the veSEQ-Castanet method.^{2,3} We have edited the manuscript to make this clearer (lines 401–403, 446–449).

We also performed PCR assays on more samples and reconfirmed the correlation between deduplicated RSV read counts and viral load (N = 141, R = 0.81, P < 2×10⁻¹⁶; figure below).

Given the inverse correlation between the number of minority variants and the number of unique reads (Figure S2). What is the justification of establishing a read depth cut off of 200x to call minority variants at any base position?

The choice of a read depth threshold is a trade-off between accuracy and sensitivity of variant calling. Using a high threshold (e.g. 1000×) can get better estimates for variant frequencies, but may miss real variants as some genomic positions do not pass the threshold and are then excluded. Using a low threshold (e.g. 10×) can account for nearly all genomic positions, but gives less accurate estimates for variant frequencies. Our choice of a read depth cut-off of 200× was based on a predefined criterion that 90% of the included samples had at least 80% of the genome passing this cut-off (see Results,

line 127). We have added the justification in Methods (lines 450–452) and now include a new Supplementary Fig. 7 (Supplementary page 10) for clarification.

Did you describe your upper limit for MAF? In Figure 1. The x-axis indicated that MAF were called from >3% - <50% is this correct? Please add this detail.

Yes, minor variants are alleles with a frequency from 3% to <50%. Any alleles with a frequency of $\geq 50\%$ are consensus alleles. We have added this detail in line 457.

Limitations when comparing the RSV genome diversity between adults and children should be included considering the cohort consists of 258 children and 9 adults. For example, on page 10 line 255 and page 12 line 320.

Thank you for pointing out this important limitation. We have added this limitation in line 321–325. We also recognise that this limitation prevents us from drawing any conclusion with regard to age, so we removed it from the abstract (lines 40–41) and our summary findings (lines 255–256).

The temporal within-host diversity is very interesting, although the authors do not describe some of the co-founders here. Median sampling of virus occurred 4 days after symptom onset and at most continued to 8 days post symptom onset. Therefore sampling over the course of any RSV infection was limited and within host diversity maybe occurring as the infection is established within the host. I recognize the

difficulties here in terms of viral detection and establishing sufficient viral load to produce sequencing read depth. However I feel this limitation should be acknowledged.

We acknowledge this limitation and have addressed it in lines 328–330.

Reviewer #2 (Remarks to the Author):

Respiratory syncytial virus (RSV) is a major cause of disease in children and the elderly. There are two main subtypes: A and B, and many different genotypes and strains. There have been studies regarding the evolution and diversity of RSV within a population, but little is known about the diversity of an RSV population within each host. Lin, et al, examine the question of RSV genetic diversity within a host. To accomplish this, the group deep sequenced the RSV population from nasopharyngeal samples collected from participants in three clinical studies of the RESCEU project. The authors identified significant differences based on patient age and virus subgroup.

The authors have a clear objectives: to study the RSV population at the host level. It has plainly described findings: that the RSV-B subgroup exhibited more diversity within-host, while RSV-A exhibited more diversity at a consensus level. The authors paid particular attention to the F gene, which codes for the fusion protein – the major target for neutralizing antibodies. The authors found the F gene to be largely genetically stable at the consensus level, much as other groups have. However, there were minor variants present in the antigenic sites, and one mutation found was in two participants had been shown in a previous study to cause a significant reduction in susceptibility to nirsevimab. Any genetic changes in this gene may have wide-ranging implications in vaccine, monoclonal antibody, and targeted small molecule development and suggests the need for continued genetic surveillance on RSV infections (particularly RSV-B) to monitor minor variants.

We thank the reviewer for this positive assessment of our work and for the helpful comments.

The authors could have made the paper more broadly accessible by explaining a bit more. For example, on page 10, line 269-273, two methods for controlling for batch effects are described but not in a way that I could understand. And a cursory search online did not yield an answer. An explanation that includes “...per sample and nucleotide diversity: (i) including batch as a regression covariate of [what] and [what]” would help. Another example is that it would strengthen the argument to mention that pairwise Manhattan distance calculation was used as a test for intrahost diversity and the pairwise patristic calculation was testing for consensus variation.

Thank you for the suggestion. We have given examples of our approaches to make them more accessible in Methods, which now read as follows (lines 488–495): “Two approaches were applied to account for batch effects on the comparisons of diversity metrics: (i) including batch as a regression covariate (e.g., regression of pairwise nucleotide diversity on sampling country, sampling season, RSV subgroup, RSV read count, participant age group, disease severity, and ‘batch’ as in Supplementary Table 2); and (ii) standardising the values within each batch to z-scores, that is, to a mean of zero and a standard deviation of 1 (e.g., Mann–Whitney U test on z-score standardised pairwise nucleotide diversity as in Fig. 2).”

We also modified the section on genetic distance to make it clearer. It now reads as follows (lines 210–215): “Within-host diversity levels between samples were compared using pairwise Manhattan distances at consensus-identical positions, where allele frequencies below the 3% threshold were converted to 0. In contrast, consensus variation between samples were compared using pairwise patristic distances, which are phylogenetic distances on RSV phylogenies.”

The authors used targeted metagenomics and used the number of unique RSV reads as a proxy for viral load to control for any variation caused by lower viral loads/unique reads. However, this method was described for influenza. Because of

the importance of this control, it would be worthwhile to show that these factors do indeed correlate with viral load so that it can be confidently used as a proxy.

We have previously performed RSV RT-qPCR and shown that RSV viral load highly correlates with the number of deduplicated RSV reads generated by the targeted metagenomic sequencing that we used in this study.¹ We have edited the manuscript to make it clearer (lines 446–449). Reviewer 1 also raised the same question, and we have provided a detailed response and more PCR data there.

The authors found that genetic diversity of RSV in older adults was higher than in infants (Fig 2b). However, can this truly be determined when the adult patient cohort was so much smaller than the infant cohort? It is an interesting finding and could perhaps lead to a new study, but I believe it is premature to conclude anything based on so few data points.

Thank you for pointing out this important limitation, which was also raised by reviewer 1. We acknowledge it is premature to conclude the difference based on only few adult participants, so we removed it from the abstract (lines 40–41) and our summary findings (lines 255–256). We also addressed this limitation in Discussion (lines 321–325).

The study found that the difference in intrahost diversity between RSV-A and RSV-B is not correlated with the duration between symptom onset and sample collection. It would be interesting to know if there is a correlation with clinical severity of the infection.

Thank you for the suggestion. We expanded our analysis and found no association between nucleotide diversity and clinical severity of the RSV infection. We added this in Methods (lines 379–386), Results (lines 206–208), and Supplementary Table 2 (Supplementary page 3, last 4 rows).

On page 12, line 314-315, the authors suggest that RSV-B is subject to greater immune pressure than RSV-A. This is an intriguing possibility that could be expounded upon.

Our hypothesis was based on a previous study on RSV, which found increased intrahost RSV diversity after a haematopoietic stem cell transplant in an immunocompromised infant with persistent RSV infection.⁴ In addition, studies have shown that RSV-B has more amino acid alterations⁵, predicted O-glycosylation site changes⁵, and indel mutations⁶ in the G gene than RSV-A, also suggesting a stronger selective pressure acting on RSV-B than RSV-A. We included these references in the manuscript to strengthen our hypothesis (lines 313–319).

Specific Issues:

There is no delineation between the Abstract and the Introduction.

We have added the heading ‘Introduction.’

p.3, l.57-58. the line sounds awkward and would flow better if it read, “...there is no efficacious antiviral for treatment or licensed vaccine to prevent RSV infection...”

Thank you for the suggestion. We have edited the sentence as you suggested.

p.3, l.69. here, and in other places, you write “antigenic sites (neutralizing epitopes)”, which makes it sound like antigenic sites are being equated to neutralizing epitopes, rather than what I believe you are saying, which is that you are speaking of neutralizing epitopes in particular.

We have edited the phrase, and it now reads as ‘antigenic sites (neutralising epitopes in particular)’ in line 69 and Fig. 1 legend.

p.5, l.139. it is unclear what the duplication rate refers to. Is it the sequential samples from the same patient? Or is it duplicate reads?

By duplication rate, we meant the ratio of duplicate reads to total RSV reads. We have rephrased this sentence to ‘... due to the differences in the ratio of duplicate reads to total

RSV reads (percent duplication rate) between batches (Supplementary Table 1).’ (lines 138–140).

l.166. The first sentence in this paragraph makes no sense until the reader gets to the end and then needs to reread it to understand it.

Thank you for pointing this out. We have moved the last sentence to the beginning, which would guide the readers to Fig. 1 and help them understand this paragraph. (lines 165–166)

l.169. One minor variant equals 0% minor variants?

In this sentence, we were talking about the percentage of participants who had minor variants encoding a nonsynonymous substitution at the antigenic sites (i.e., potential antigenic variants) each year. Thus, the figures referred to the percentage of participants instead of minor allele frequency. Thank you for the question. We have rephrased this sentence to avoid any confusion (lines 169–172).

l.172. “Two of these minor covariants coexist in the same sample...substitution from Ile to Thr.” Are the two covariants in one codon that leads to the AA change, or distant from each other and only one of them causes an AA change?

As we showed in Table 2, these two substitutions were located on the same codon (nucleotide positions 782 and 783), causing the amino acid change. We added this information to the sentence to avoid any doubt (line 173–175).

l.188. Are “pairwise nucleotides” neighboring nucleotides or anywhere in the genome?

Pairwise nucleotide diversity refers to the nucleotide differences in the ‘same’ genomic position of a ‘pair’ of samples.⁷ We have provided the formula for calculating pairwise nucleotide diversity and explained the meaning of each symbol in the formula (paragraph after line 457).

L195. “The L gene had significantly higher nucleotide diversity than...” Is that because the L gene is the largest gene? Or is this based on frequency within the gene?

Pairwise nucleotide diversity accounts for the length of a genetic sequence. Specifically, the length of a genetic sequence is the denominator in the calculation of pairwise nucleotide diversity. Therefore the fact that the L gene had greatest nucleotide diversity is based on the proportion of pairwise nucleotide differences within the gene instead of its length. We have added this in the Results (lines 197–199).

Supplementary figures 1, 2, and 3 describe the results in the figure legend. They should only describe the figure if the results are described in the body.

We have removed the description of the results from the figure legends in Supplementary Figs. 1, 2, and 3.

Reviewer #3 (Remarks to the Author):

This study examines intra-host diversity of human respiratory syncytial virus (RSV). Using deep-sequencing of nasopharyngeal swabs from 3 countries during consecutive RSV seasons, the authors find that intra-host diversity differs between the two RSV subgroups, inversely from their population level diversity. Longitudinal samples of a subset of the infected infants showed mostly nonlinear changes over time in diversity.

The data are interesting and are well-presented. Intra-host diversity of acute RNA viral infections in general remains an understudied field. This manuscript will enhance our understanding of these infections.

We appreciate the reviewer’s positive feedback and valuable suggestions which help us improve our manuscript significantly.

There are a few minor concerns whose clarification would enhance the manuscript.

1. Figure 1. Analysis of G diversity provided as a counterpoint to F would be beneficial as G is also an antigen and has previously been reported to have hypermutation at the population level in the C-terminus.

Our analysis focused on the F protein as it has been a target of several promising vaccines and monoclonal antibodies. However, we agree that within-host diversity of the G gene would also be of great interest and could act as a counterpoint to the F gene. We expanded our analysis to the G gene and found that the median frequency of minor variants was higher in the G gene than in the F gene. The 5' end of the G gene (i.e., mucin-like region II or 2nd hypervariable region) had similar median frequency of minor variants to that of all potential antigenic sites of the G gene. We have added this part in Introduction (lines 81–88) and Results (lines 179–187 and page 28 Fig. 1b).

2. Figure 2. It is not clear how much the analysis of older adults contributes to the impact of the overall data set given the large disparity in sample sizes between that population and infants. A focus on infant samples would mitigate the confounding factor of prior immunity.

We acknowledge the scarcity of older adult samples (7 older adult samples and 137 infant samples), which prevents us from drawing any conclusion between age groups. We removed the finding on the difference in within-host diversity between older adults and infants from the abstract (lines 40–41) and our summary findings (lines 255–256), and addressed this limitation in the Discussion (lines 321–325).

3. Figure 3. The text states that 34 participants had multiple samples collected daily but only 12 individuals are shown. If the samples for 22 participants were excluded from the analysis due to sequencing in different batches, that would be important to note in the text.

In Fig. 3, we only included participants who had *more than* two samples collected. Three participants whose samples were sequenced in different batches and 19 participants who

had only two samples collected were excluded here. We have added this information to the figure legend for clarity.

4. Figure S2: There appear to be many more minor variants in Batch 2 compared to the other batches, even adjusting for overall number of unique reads.

Overall, samples in batch 2 had a higher mean unadjusted minor allele frequency (MAF) per sample than batches 3 and 4, as we have shown in Supplementary Fig. 1a, necessitating the use of models that accounted for batch as a confounder (Supplementary Table 2). We also mentioned in the main text that when sampling fraction is small (i.e., few unique reads were sequenced), there is a greater variance of MAF (lines 148–149).² Therefore, the higher mean MAF per sample in batch 2, together with a greater variance of MAF in low-burden samples (i.e., those with $\leq 4.5 \log_{10}$ uniquely mapped reads), caused a greater number of minor variants in some low-burden samples in batch 2. There were 25 low-burden samples with >10 minor variants in batch 2. The samples with the greatest numbers of minor variants in this batch represented both subgroups, with 11 RSV-A and 14 RSV-B, contributing similarly to diversity in both subgroups. We have added this clarification to Supplementary Fig. 2 legend (Supplementary page 5).

References

1. Lin GL, Golubchik T, Drysdale S, et al. Simultaneous viral whole-genome sequencing and differential expression profiling in respiratory syncytial virus infection of infants. *J Infect Dis* 2020;222:S666-S71.
2. Lythgoe KA, Hall M, Ferretti L, et al. SARS-CoV-2 within-host diversity and transmission. *Science* 2021;372:eabg0821.
3. Bonsall D, Golubchik T, de Cesare M, et al. A comprehensive genomics solution for HIV surveillance and clinical monitoring in low income settings. *J Clin Microbiol* 2020;58:e00382-20.
4. Grad YH, Newman R, Zody M, et al. Within-host whole-genome deep sequencing and diversity analysis of human respiratory syncytial virus infection reveals dynamics of genomic diversity in the absence and presence of immune pressure. *J Virol* 2014;88:7286-93.

5. Matheson JW, Rich FJ, Cohet C, et al. Distinct patterns of evolution between respiratory syncytial virus subgroups A and B from New Zealand isolates collected over thirty-seven years. *J Med Virol* 2006;78:1354-64.
6. Schobel SA, Stucker KM, Moore ML, et al. Respiratory syncytial virus whole-genome sequencing identifies convergent evolution of sequence duplication in the C-terminus of the G gene. *Sci Rep* 2016;6:26311.
7. Nelson CW, Hughes AL. Within-host nucleotide diversity of virus populations: insights from next-generation sequencing. *Infect Genet Evol* 2015;30:1-7.

REVIEWERS' COMMENTS

Reviewer #1 (Remarks to the Author):

Thank you to the authors for their detailed response to my comments. They have clarified my comments and amended the manuscript well.

Reviewer #2 (Remarks to the Author):

Lin et al. have assessed respiratory syncytial virus (RSV) genome sequences from a large cohort of European children and a small number of adults infected by RSV for subgroup genome diversity and for diversity among individuals. They have responded well to the suggestions of the previous reviews. Only a few problems remain.

Inclusion of the CCD, a region known to be conserved, with the mucin-like region I (I.186) would likely artificially reduce the frequency of minor variants that would survive in the individual. In other words, if the CCD were removed from MLP-1, the % variants MLR-1 and MLR-2 would likely be closer. That would be a more fair/accurate way to make this comparison.

I.198. If "pairwise" means for any particular nucleotide position, then by definition the number would not be related to the length of the gene. Stating this would be much clearer than avoiding the issue by using "presumably".

I.259. Are these differences in F antigenic sites novel, or have they been reported before but do not affect neutralizing antibodies?

I.260. What does "stochastic" mean here?

I.288. HBV is not an RNA virus. It is a DNA virus that has an RNA step in its life-cycle. Perhaps us "persistent viruses with an RNA step in their life-cycle".

I.314. Antibody selected mutations would be limited to F and G, and therefore would likely be different from genome-wide substitution rates. Is that true here? In any case, this distinction should be made.

I. 342. I64I is not a mutation. Is this nomenclature (I64I/T) meant to indicate a mixed population of viruses at I64 of I or T? Same question for K68K/E. In any case, the mutations would only be I64T and K68E.

Other.

I.48. "Variable" compared to what?

I.64. Negative-sense single-strand (Important to say negative sense)

I.84. ...has been shown to be a target...

I.96. "Immunocompromised" is more important than stem cell treated and could be mentioned first to set the stage for the reader.

I.146. Confusing because the second half of the sentence accounts for part (0 variants) of the first part. Two sentences would make it clearer.

I.176. "reported" would be more accurate than "demonstrated"

I.184. ...gene in the virus genome.

I.302. ...antibodies had been...

Reviewer #3 (Remarks to the Author):

The authors have addressed all of my previous concerns.

Reviewer #2:

Lin et al. have assessed respiratory syncytial virus (RSV) genome sequences from a large cohort of European children and a small number of adults infected by RSV for subgroup genome diversity and for diversity among individuals. They have responded well to the suggestions of the previous reviews. Only a few problems remain.

We thank Reviewer #2 for further comments, which we have used to improve the manuscript.

Inclusion of the CCD, a region known to be conserved, with the mucin-like region I (I.184) would likely artificially reduce the frequency of minor variants that would survive in the individual. In other words, if the CCD were removed from MLP-1, the % variants MLR-1 and MLR-2 would likely be closer. That would be a more fair/accurate way to make this comparison.

We agree that minor variants in the conserved central domain (CCD) tended to have lower frequencies and should have been separated in this comparison. We have revised this section to show separately the median minor variant frequencies in these three regions of the G gene (i.e., CCD and mucin-like regions I and II) and the statistical test of the differences. (lines 185–188)

l.199. If "pairwise" means for any particular nucleotide position, then by definition the number would not be related to the length of the gene. Stating this would be much clearer than avoiding the issue by using "presumably".

We have edited this sentence to "These significant differences were by definition due to the mean proportion of pairwise nucleotide differences at each genomic position within the L gene instead of the length of the L gene." (lines 198–200)

l.257. Are these differences in F antigenic sites novel, or have they been reported before but do not affect neutralizing antibodies?

These minor variants in the F antigenic sites are novel except for S255N¹, but its susceptibility to monoclonal antibodies has not been examined. (lines 258–259)

l.259. What does "stochastic" mean here?

By stochastic, we meant random here. To make it clearer, we added random in brackets. (line 260)

1.288. HBV is not an RNA virus. It is a DNA virus that has an RNA step in its life-cycle. Perhaps us “persistent viruses with an RNA step in their life-cycle”.

Thank you for pointing this out. We have decided to separate the RNA and DNA viruses to make it clearer. (lines 288–289)

1.315. Antibody selected mutations would be limited to F and G, and therefore would likely be different from genome-wide substitution rates. Is that true here? In any case, this distinction should be made.

Selective pressure from antibodies is limited to F and G, but selective pressure caused by other host immune responses (e.g., innate immunity or T cell-mediated cytotoxicity) can act on other parts of the genome. We have addressed the possible sources of selective pressure in the text. (lines 315–316)

1. 343. I64I is not a mutation. Is this nomenclature (I64I/T) meant to indicate a mixed population of viruses at I64 of I or T? Same question for K68K/E. In any case, the mutations would only be I64T and K68E.

Yes, I64I/T and K68K/E meant a mixed population of viruses with some variants not carrying the mutations at these sites. We have changed the nomenclature to I64T and K68E to represent the actual mutations instead of mixed viral populations. (line 343)

Other.

1.47. “Variable” compared to what?

We meant ‘different’ prevalence of monoclonal antibody-escape mutants between the two subgroups. We have changed the word from variable to different. (line 47)

1.63. Negative-sense single-strand (Important to say negative sense)

This is a key point; we have added it. (line 63)

1.83. ...has been shown to be a target...

Thank you, we have changed it. (line 83)

1.95. “Immunocompromised” is more important than stem cell treated and could be mentioned first to set the stage for the reader.

We have reordered the sentence. It reads now: within-host RSV diversity increased in an immunocompromised infant with persistent RSV infection following a haematopoietic stem cell transplant. (lines 94–96)

l.146. Confusing because the second half of the sentence accounts for part (0 variants) of the first part. Two sentences would make it clearer.

We have separated it to two sentences. (lines 146–147)

l.175. “reported” would be more accurate than “demonstrated”

We have changed the word according to your suggestion. (line 175)

l.182. ...gene in the virus genome.

We have edited the sentence. (line 182)

l.303. ...antibodies had been...

Thank you for the suggestion. We have changed it to the present tense as this sentence refers to the current situation. We have also restructured the sentence to make it clearer. It reads now: It is important to identify neutralisation escape mutants in immunoprophylaxis-naïve children in the era before RSV monoclonal antibodies become extensively used. (lines 301–303)

Reference

1. Tabor DE, Fernandes F, Langedijk AC, et al. Global molecular epidemiology of respiratory syncytial virus from the 2017-2018 INFORM-RSV study. *J Clin Microbiol* 2020;59:e01828-20.